# Morphological Changes in Storm Hinnamnor and the Numerical Modeling of Overwash

**Bohyeon Hwang** [1], **Kideok Do** [2,*] **and Sungyeol Chang** [3]

1   Department of Convergence Study on the Ocean Science and Technology, Korea Maritime and Ocean University, Busan 49112, Republic of Korea; zpzpxpxp@gmail.com
2   Department of Ocean Engineering, Korea Maritime and Ocean University, Busan 49112, Republic of Korea
3   Haeyeon Engineering and Consultants Corporation, Gangneung-si 25623, Republic of Korea; sungyeoly@naver.com
*   Correspondence: kddo@kmou.ac.kr; Tel.: +82-51-410-5248

**Abstract:** Constant changes occur in coastal areas over different timescales, requiring observation and modeling. Specifically, modeling morphological changes resulting from short-term events, such as storms, is of great importance in coastal management. Parameter calibration is necessary to achieve more accurate simulations of process-based models that focus on specific locations and event characteristics. In this study, the XBeach depth-averaged model was adopted to simulate subaerial data pre- and post-storms, and overwash phenomena were observed using the data acquired through unmanned aerial vehicles. The parameters used for the model calibration included those proposed in previous studies. However, an emphasis was placed on calibrating the parameters related to sediment transport that were directly associated with overwash and deposition. Specifically, the parameters corresponding to the waveform parameters, wave skewness, and wave asymmetry were either integrated or separated to enable an adequate representation of the deposition resulting from overwash events. The performance and sensitivity of the model to changes in volume were assessed. Overall, the waveform parameters exhibit significant sensitivity to volume changes, forming the basis for calibrating the deposition effects caused by overwashing. These results are expected to assist in the more effective selection and calibration of parameters for simulating sediment deposition due to overwash events.

**Keywords:** morphological response; UAV (unmanned aerial vehicle); overwash; numerical modeling; XBeach

## 1. Introduction

Changes in coastal areas occur over various timescales and can be attributed to both human activities and natural phenomena. Among these, short-term extreme events, such as storms, can cause significant morphological changes in coastal areas. These events can lead to erosion and even overwash, which can directly affect not only the beach, but also human settlements located beyond revetments. Hence, precise observations and effective numerical modeling must be performed for the management of erosion and the associated morphological changes induced by overwashing along coastlines [1]. Various models can be used to perform such a numerical modeling; however, they often involve different assumptions and empirical formulations [2]. To achieve reasonable results, the parameters provided by these models must be calibrated according to the specific region and target event or phenomenon [3].

In 2022, Storm Hinnamnor significantly affected Songjeong Beach in Busan, South Korea. To assess this impact, a survey was conducted using unmanned aerial vehicles (UAVs), which revealed significant erosion at the front of the beach and deposition due to overwashing at the back. In this study, the XBeach model [2], which is effective for simulating erosion caused by storms on sandy beaches [4–6], was employed to simulate

both erosion and overwash-induced deposition. XBeach was originally developed to simulate the collapse of barrier islands on dissipative beaches, and its default parameters have been criticized for overestimating offshore sediment transport [7]. Because of the model's inherent nonlinearity and its reliance on empirical formulations, a proper parameter calibration is essential for obtaining accurate results. Various calibration methods have been proposed in previous studies [4,5,8]. However, the trial-and-error method, which typically relies on the experience and knowledge of model users, is commonly used. For an effective calibration, the roles and sensitivities of the parameters within the model must be understood.

This study aims to simulate the overwash and resultant deposition during a storm period by calibrating the waveform parameters in XBeach. XBeach utilizes the advection–diffusion equation for sediment transport. In this equation, the non-linearity term includes the effects of wave skewness and wave asymmetry. The influence of wave skewness and asymmetry on the sediment advection velocity has been well explained by van Thiel de Vries [9]. In XBeach, the parameters for the wave asymmetry and skewness are defined as facAs and facSk, respectively, and can be represented as facua when these parameters have identical values. Many prior studies have opted for simulations with facua instead of facAs and facSk to reduce the number of parameters requiring calibrations, and such simulations have shown high accuracy for erosion. Facua is often directly linked to asymmetric flow, with Nederhoff [10] demonstrating a high accuracy for erosion (BSS of 0.83) by calibrating facua to a value of 0.25 and explaining that an increase in facua reduces the net sediment transport in the offshore direction. Saber et al. [11] discussed the significant impact of facua on the onshore (offshore) velocity, which is closely linked to sediment transport, stating that the default value of facua (0.1) induced an overestimation of erosion. They suggested a more effective simulation of erosion by correlating it with the average slope angle. In their study of multiple storm events in XBeach simulations, Jin et al. showed that the sensitivity of facua was the highest, significantly influencing the model's accuracy [5]. Therefore, this research aims to compare and analyze the results of separating facua into facAs and facSk, not only in terms of the erosion caused by overwash, but also for the resultant deposition.

In this study, two statistical schemes and a volume acquired through a UAV observation are employed as the basis for evaluating the model's results. While the beach data for pre- and post-storm events were obtained, depth data immediately after the storm were not available; hence, the evaluation was restricted to the beach data.

The research area and UAV survey results are presented in Section 2. Section 3 describes the XBeach model, input data required for the model, and parameters. It also introduces the statistical schemes used for the model accuracy assessment. Section 4 presents the results, and Sections 5 and 6 present the discussion and conclusions, respectively.

## 2. Field Data Collection

### 2.1. Study Area

Figure 1 shows the location of Songjeong Beach in the study area. Songjeong Beach is a predominant pocket beach located in Busan, Republic of Korea, characterized by a wave-dominated area (129°11′45″–129°12′23″ E, 35°10′24″–35°10′56″ N). With a length of approximately 1.2 km, the beach width ranges from 43 to 60 m. The beach primarily consists of sand with a median grain size ($D_{50}$) of 0.42 mm. In addition, structures composed of boulders were observed on the flank of the T headland. This region has been consistently subjected to erosion, prompting the Ministry of Oceans and Fisheries to categorize it with erosion grades C to D for ongoing management (A = excellent, B = moderate, C = concern, and D = serious). Consequently, annual nutrition projects are currently underway in this area. Figure 1 shows the presence of a well-defined crescentic sandbar prior to a storm event. While the typical tidal range is approximately 1–2 m, a notable elevation in the sea level occurs during a storm surge.

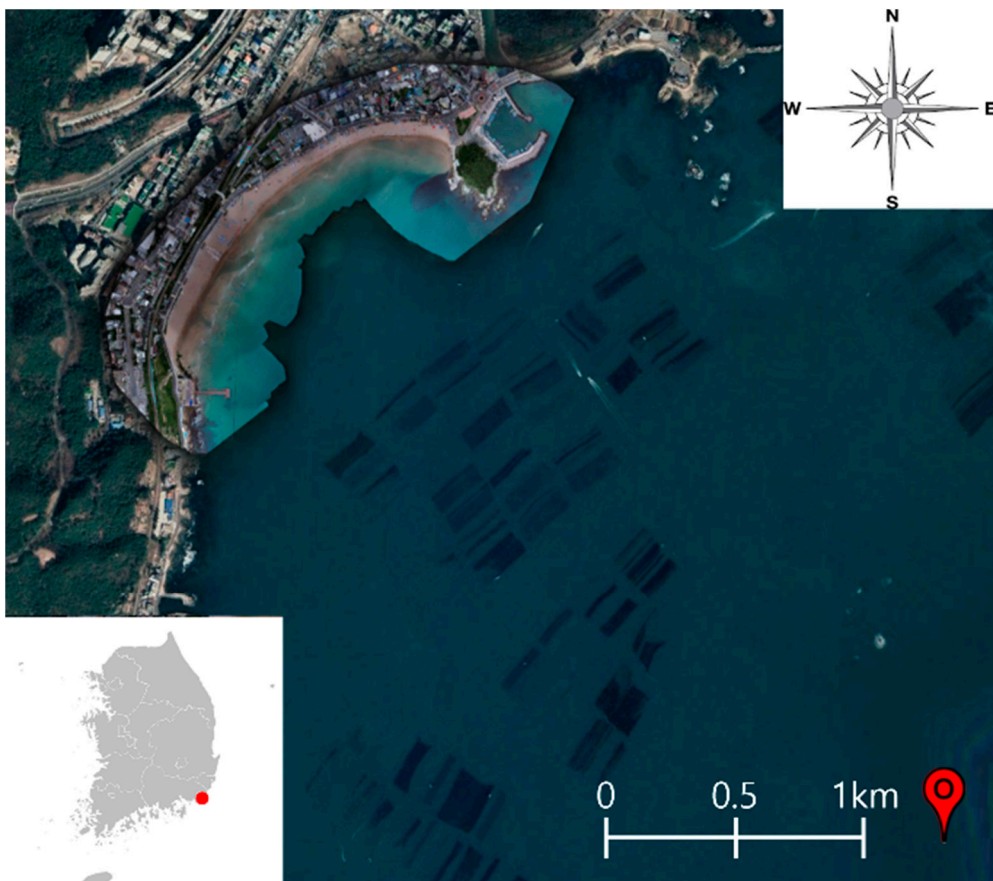

**Figure 1.** Location of Songjeong beach. Google Earth with orthomosaic image from the UAV, taken on 29 August 2022. The buoy was installed at 'O'.

## 2.2. Storm Condition

On 28 August 2022, Storm Hinnamnor originated in the northwestern Pacific and exhibited an initial wind speed of 15 m/s as it progressed northward. The central minimum pressure reached 955 hPa and struck Busan, Republic of Korea, on 5 September at 21:00. The wave data were obtained using a buoy deployed by the Korea Hydrographic and Oceanographic Agency (KHOA). Figure 2a shows the significant wave height, wave period, wave direction, and tide level over time during the storm. The maximum recorded significant wave height was 10.5 m, with the wave period ranging from 9 to 10 s. Using this dataset, the wave rose diagram (Figure 2b) shows easterly waves at 90–100° during the storm period. Figure 2c shows the path of Storm Hinnamnor throughout its formation and dissipation phases. The storm had a direct impact on Busan, leading to a low central pressure and subsequent storm surge, resulting in a rise in sea level, causing erosion and overwashing with substantial waves.

## 2.3. Observation Method

Morphological changes in the beach due to Storm Hinnamnor were captured through observations conducted with a UAV, both pre- and post-storms. Observations were performed on 29 August 2022, prior to the storm strike, and on 7 September 2022, following its impact. A Phantom 4 RTK was utilized for these observations, set to operate at an altitude of 100 m with an overlap and side lap of 80% and a speed of 7.9 m/s. Approximately 720 images were captured. Equipped with real-time kinematic (RTK) technology, the Phantom 4 RTK allows the production of highly accurate data through a real-time correction using the Global Navigation Satellite System (GNSS). Nonetheless, for the accuracy assessment and calibration, 12 ground control points (GCPs) were established; their locations are

shown in Figure 3. The integration of approximately 720 photographs and data calibration with the GCPs were performed using the Pix4Dmapper (ver.4.8.4) software. This process enabled the acquisition of high-accuracy values with an accuracy assessment based on the observational data and GCPs presented in Table 1.

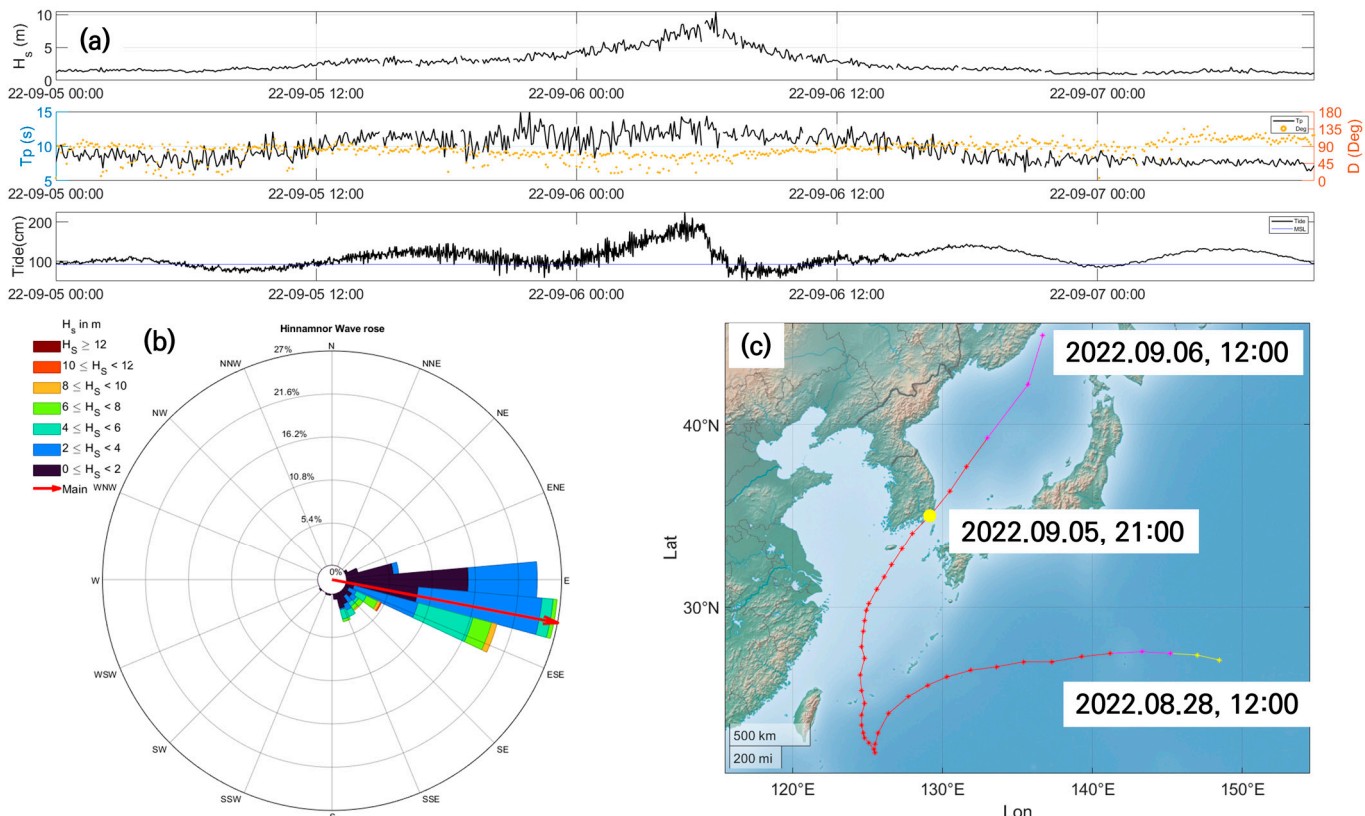

**Figure 2.** Wave data during the storm period. (**a**) Significant wave height, spectral peak period, wave direction, and tide elevation time series at Songjeong Beach. (**b**) Wave rose diagram. (**c**) Paths of storm Hinnamnor with the study area (yellow dot) and arrival time. The color of the lines indicates the intensity of the storm. In the depicted graph, the yellow line represents TS (Tropical Storm), the pink line denotes STS (Severe Tropical Storm), and the red line signifies TY (Typhoon, Storm).

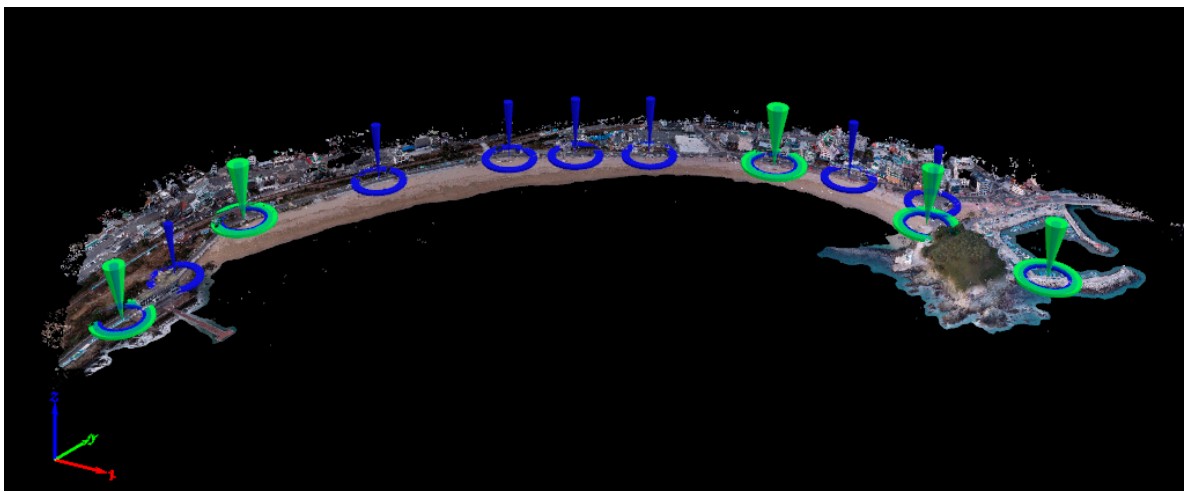

**Figure 3.** Positions of the ground control points (GCPs) in Songjeong Beach.

**Table 1.** Standard deviation (Std.Dev) and root mean square error (RMSE) per observation ID (Obs ID).

| Obs ID | Std. Dev (X/Y/Z) (m) | RMSE (X/Y/Z) (m) |
|---|---|---|
| Pre-storm | 0.005/0.006/0.013 | 0.005/0.005/0.013 |
| Post-storm | 0.007/0.001/0.009 | 0.007/0.011/0.009 |
| Recovery | 0.001/0.009/0.010 | 0.010/0.009/0.010 |

*2.4. Volumetric Changes and Classification of the Beach*

UAVs were used to examine the changes in the beach before and after a storm event. Table 2 presents the volumetric analysis and the calculated erosion quantities for Songjeong Beach derived from the UAV survey data. The vertical reference for the volume measurement was based on the elevation calculated using the approximate lowest water level and datum level (App.LLW). Volume comparisons were conducted in common areas where the pre- and post-storm regions overlapped to ensure consistency. This approach was necessary owing to the lack of altitude observations by UAVs in some areas caused by the run-up from storm surges. Prior to the arrival of the storm on 29 August 2022, the observed volume was 69,939.2 m$^3$. The post-storm assessment on 7 September 2022 indicated an erosion value of 9533 m$^3$, resulting in a remaining volume of 60,406.2 m$^3$. Figure 4a depicts the elevation differences post- and pre-storm events based on the UAV data comparison. Erosion is evident along the frontal section of the coastline, with overwash and deposition observed in the rear section near the revetment site. The most extreme erosion event occurred in the northeastern SW region of the beach.

**Table 2.** Erosion and volume of pre- and post-storm events.

| | Pre-Storm | Post-Storm |
|---|---|---|
| Obs Date | 29 August 2022 | 7 September 2022 |
| Volume (m$^3$) | 69,939.2 | 60,406.2 |
| Erosion (m$^3$) | 9533 | |

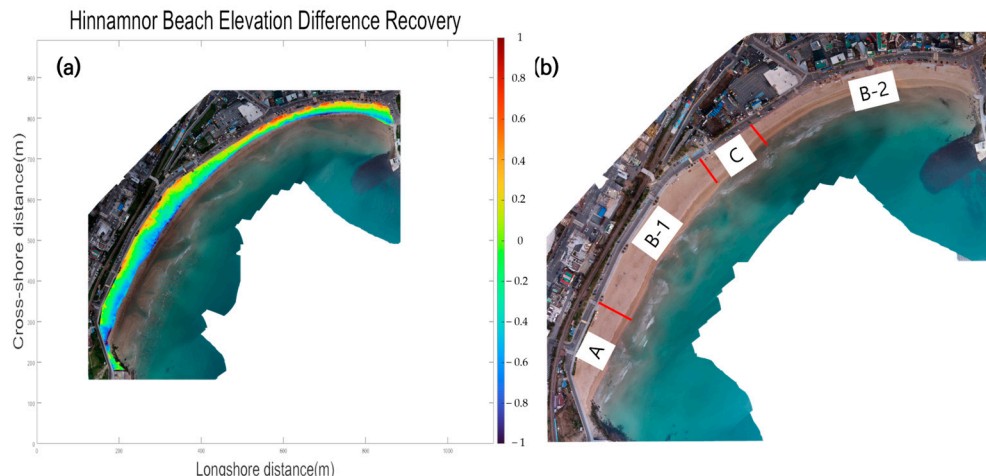

**Figure 4.** (**a**) Elevation differences after and before the arrival of Storm Hinnamnor. (**b**) Beach divided according to their characteristics.

Based on the elevation differences shown in Figure 4a, the beaches were categorized into three types. Type A refers to areas with significant erosion throughout; type B refers to areas with erosion at the front and deposition in the rear owing to overwash; and type C refers to areas with erosion at the front and minimal changes in the rear. The results when these areas are marked in the study area are shown in Figure 4b. Type A is located in the SW region of the beach, and type B is the predominant type.

## 3. Method of Numerical Modeling

### 3.1. Description and Domain with Model Set

XBeach is a two-dimensional, depth-averaged (i.e., 2DH) numerical model that integrates fluid dynamics and morphodynamic processes to simulate wave propagation and morphological changes under short-term storm wave conditions. The storm event in the study area was modeled using XBeach (1.23.5527) and the surfbeat mode (XBSB), specifically designed to simulate hydrodynamics and morphological changes in narrow areas, such as beaches, dunes, and barrier islands, during storm wave events. The XBSB model calculates the hydrodynamics (shortwave, longwave, and roller energy) with wave and water-level inputs as boundary conditions. Subsequently, it simulates the morphodynamics, including sediment transport and the resulting bed-level changes. Figure 5a,b show the orthogonal grid and pre-depth data used for the XBeach modeling. Depth data were provided by an external agency and utilized in this study. The data surveyed on 23 August 2022, before the storm, underwent precise calibrations with an error margin of approximately 1 m. Depth measurements were conducted from 0 to 5 m using a single beam, whereas depths beyond this range were surveyed using a multibeam approach. Owing to the essential nature of data correction, a calibration was performed using DGNSS, a motion sensor, and a gyrocompass. Additionally, tidal and sound speed corrections were implemented to minimize the data errors. The grid resolution was configured to increase the density closer to the nearshore, thereby optimizing the simulation of the morphological changes within the surf and swash zones. Conversely, the grid size was proportionally enlarged seaward. Specifically, the cross-shore grid size ranged from 3 to 20 m, while the longshore grid maintained a consistent resolution of 10 m, resulting in a 201 × 201 grid configuration. The structures, including revetments, were designated as non-erodible layers. The wave boundary conditions were defined using the time-varying JONSWAP spectrum based on the observation data, incorporating the parameters (Hs, Tp, and Dp) outlined in Section 2.2. Temporal variations in the water level were derived from the tide data in Section 2.2, serving as the water level boundary conditions. The modeling period spanned from 29 August to 7 September, following the UAV observational data. For the computational efficiency, significant wave heights of less than 1 m were excluded. The morphological acceleration factor (morfac) was set at 10.

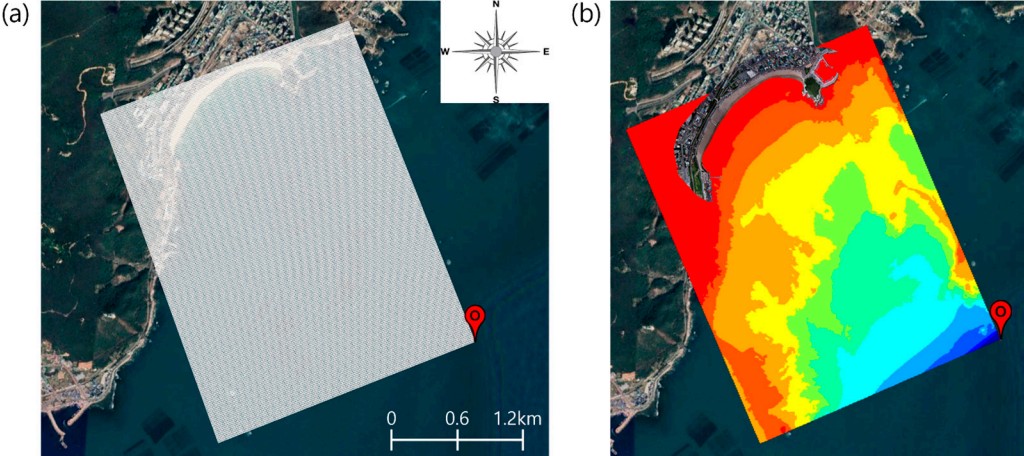

**Figure 5.** (**a**) Orthogonal grid domain for XBeach modeling with Google Earth image. (**b**) Depth with UAV orthomosaic and Google Earth image. The red dots represent the locations where wave data were acquired.

### 3.2. Parameter Calibration

The default parameter settings of XBeach were calibrated for North Sea wave conditions off the Dutch coast and were specifically designed to simulate the collapse of barrier islands during storms. Consequently, the offshore sediment transport and morphological

changes were overestimated under typical beach conditions. To address this limitation, ongoing discussions focus on various parameter calibration methods [3,4,12,13]. In this study, a trial-and-error approach was employed based on the parameters proposed in previous studies [5] along with additional parameters, with the aim of simulating overwash during storm events.

Various formulations were embedded in the XBeach model, allowing for an effective simulation of the morphological changes specific to the study area. Among these, the discussions on sediment transport equations are extensive. XBeach offers sediment transport formulations from Soulsby–Van Rijn [14], van Thiel de Vries–Van Rijn [9,15,16], and Van Rijn [17]. A primary distinction between the widely used Soulsby and van Thiel de Vries–Van Rijn equations is that Soulsby employs a drag coefficient to determine the equilibrium sediment concentration, which is absent from van Thiel de Vries–Van Rijn. Additionally, the van Thiel de Vries–Van Rijn equation distinguishes between currents and waves in its calculation of critical velocity, whereas Soulsby does not. Although each formulation has its strengths, the effectiveness largely depends on the features of the coastal environment. For instance, De Vet et al. compared the Soulsby–Van Rijn and van Thiel de Vries–Van Rijn formulations, suggesting that the latter provided more credible results [18]. The van Thiel de Vries–Van Rijn equation was noted to represent breaching more effectively and overwashing than the Soulsby equation, which overestimated the erosion rates. Conversely, Craig et al. [19] favored Soulsby in modeling the microtidal, wave-dominated hydrodynamic environment of the Upper Texas Coast (UTC), effectively simulating phenomena, such as collision, overwash, and inundation. Similarly, Orzech et al. [20] showed commendable Brier Skill Score results for Monterey Bay using the XBeach 2DH model. Monterey Bay, located in California, is characterized by its mild alongshore currents, rip channel bathymetry, and tidal range of approximately 1.6 m, features that resonate closely with the Songjeong Beach studied in this research. Consequently, this study aimed to simulate sediment transport by employing the Soulsby equation.

Within the XBeach model, two formulations related to shortwave breaking exist: the Roelvink et al. and Daly formulas. The Roelvink equation was used by default for all the statistical analyses. This equation is based on an empirical formula that incorporates the breaking coefficient (gamma) and the ratio of the wave height to the water depth. However, Daly et al. observed that this equation tended to underestimate the wave energy dissipation when the water depth increases rapidly. In response to this result, Daly introduced the parameter gamma2, which allowed for a more precise determination of wave breaking, even in areas with sharp changes in the water depth. Furthermore, when dealing with depths that have a plane slope instead of steep inclines, the differences in the results generated by the two formulas are negligible. In this study, based on the previous research, specific parameters and their associated formulas were adopted, and the Daly equation was used for the model setup.

Based on this, the gamma and gamma2 parameters were employed, and the waveform parameters related to the wave skewness and asymmetry, which were closely related to sediment transport, were calibrated. XBeach offered the two waveform equations provided by Ruessink et al. [21] and van Thiel [9]. Both computed skewness and asymmetry. The formula of Ruessink et al. suggests the Ursell number parameter, which is based on over 30,000 field observations of orbital skewness and asymmetry collected under non-breaking and breaking conditions. Based on the significant wave height, wave period, and water depth, they reported that the skewness and asymmetry were efficiently represented. Another approach involved the equation proposed by van Thiel et al., which was an extension of the wave-shaped model presented by Rienecker et al. [22]. This model describes the shortwave shape as a weighted sum of eight sine and cosine functions. Skewness and asymmetry were then represented based on the computed near-bed shortwave flow velocity using this model. However, it remains unclear which of the two equations is more accurate [11]. In this study, the default equation provided by van Thiel was used.

Several studies discuss the effects of skewness and asymmetry on sediment transport [11,13,23]. According to van Thiel et al. [9], Stokes waves exhibit higher onshore flow velocities than offshore waves. This implies that the sediment movement is more pronounced during the wave crest than during the wave trough. Consequently, skewness and asymmetry influence the Eulerian velocities through an additional onshore sediment advection velocity, denoted as $u_a$. Using skewness and asymmetry, $u_a$ can be represented as:

$$u_a = (f_{Sk}S_k - f_{As}A_s)u_{rms} \tag{1}$$

where $f_{Sk}$ is a coefficient related to the phase shift of the intrawave sediment concentration, flow, and skewness. $f_{As}$ is the coefficient pertaining to the phase shift between the flow and the suspended sediment for asymmetry. Assuming that $f_{Sk}$ and $f_{As}$ are equivalent, they can be expressed as $f_{ua}$, and the equation can be summarized as:

$$u_a = f_{ua}(S_k - A_s)u_{rms} \tag{2}$$

In XBeach, facua is denoted as a coefficient associated with the phase shift between intrawave sediment suspensions and orbital flow. In previous studies [5], facua was used. A comparison with the sets of facSk and facAs was conducted in this study. Based on these observations, it was suggested that the profile shapes in the shoaling and breaker zones were significantly influenced by facSk. An increase in fasSk was linked to an increase in wave skewness, which was associated with an increase in offshore sediment fluxes, whereas [24] the cross-shore profiles in the surf and swash zones were believed to be affected by facAs. As facAs increases, the asymmetry in the waves is enhanced, leading to an increase in onshore sediment transport. Based on this, the set parameter values and their ranges have been summarized in Table 3.

**Table 3.** Parameter descriptions and values.

|  | Description | Calibration Value or Range |
|---|---|---|
| gamma | Breaker parameter in Daly formula | 0.52 |
| gamma2 | Set stop point of breaking in Daly formula | 0.3 |
| facua | Time-averaged flows due to wave skewness and asymmetry (facua = facSk = facAs) | 0.09:0.03:0.42 |
| facAs | Time-averaged flows due to wave asymmetry | 0.09:0.03:0.36 |
| facSk | Time-averaged flows due to wave skewness | 0.09:0.03:0.36 |
| $D_{50}$ | Median grain size of sediment (mm) | 0.42 |
| break | Type of wave breaking formula | Roevink_Daly |
| form | Type of sediment transport formula | Soulsby-vanrijn |

Observational UAV data were used to calibrate the parameters. Comparisons between the observational and model values were only performed in overlapping areas, which, owing to the characteristics of the UAV, were confined to the subaerial region [7,19]. Based on this, the simulation skill proposed by Gallagher et al. (1998) [25] could be employed to assess the accuracy of the elevation difference obtained from the UAV and model output. This skill is defined as follows:

$$Skill = 1 - \frac{\sum_{i=1}^{N}\left(dz_{b_{UAV,i}} - dz_{b_{XBeach,i}}\right)^2}{\sum_{i=1}^{N}\left(dz_{b_{UAV,i}}\right)^2} \tag{3}$$

where $N$ is the number of data points (i.e., the number of grids) in the overlapping section between the UAV's pre- and post-data and the output of the model. $dz_{b_{UAV,i}}$ denotes the observed bed-level change in $i$, whereas $dz_{b_{XBeach,i}}$ is the modeled bed-level change at point $i$. A skill value of 1 indicated a perfect match between the model predictions and observed data, indicating optimal accuracy. A skill of 0 suggested that the model's accuracy was

equivalent to random or base-level predictions, whereas a negative skill indicated that the model's predictions were less accurate and that it performed poorly (Table 4). Furthermore, the determination of the mean error allowed us to distinguish between biases resulting from systematic differences in the model outcomes and random variations. The equations for this are as follows:

$$Bias = \frac{1}{N}\sum_{i=1}^{N}\left(z_{b_{post-storm,Model,i}} - z_{b_{post-storm,UAV,i}}\right) \tag{4}$$

where $z_{b_{post-storm,Model,i}}$ is the post-elevation of the model in cell $i$ and $z_{b_{post-storm,UAV,i}}$ is the post-elevation of the observation data. A positive bias indicated that the model predicted higher elevations than those observed, whereas a negative bias signified that the model predicted lower elevations than those observed (Table 4).

**Table 4.** Qualification of XBeach's performance.

|  | **Skill** | **Bias** |
| --- | --- | --- |
| >0 | Good (=1, perfect) | Model predicts higher results than Obs |
| =0 | Nothing | Same |
| <0 | Poor | Model predicts lower results than Obs |

## 4. Results

### 4.1. Parameter Calibration Results: Sensitivity of Volumetric Information

In this study, as previously mentioned, gamma and gamma2 related to wave breaking were utilized according to the values used in the previous research by Jin et al. [5]. Only the waveform parameters, namely, facua, facAs, and facSk, were employed for calibrations. Additionally, the performance of the model was evaluated based on two statistical schemes and observed volumetric changes in the beach. The sensitivity of each parameter to the volume was first examined. As shown in Table 3, 12 facua cases were selected to simulate beach morphological changes. Figure 6 shows the changes in the volume with increasing facua values. The volume increased almost proportionally with the increase in facua [26], reaffirming the significant impact of wave nonlinearity on sediment movement. However, as depicted in the graph of facua in Figure 6, although the volume increases uniformly with an increase in facua, the skill value, which represents the accuracy, decreases as it approaches the observed volume value of 60,406.2 m$^3$. This suggests that the increase in facua is unsuitable for simulating the unbalanced phenomena of erosion at the front and deposition at the rear because it leads to an overall increase in volume across the entire beach area.

The calibration was conducted by separating facua into facAs and facSk. A total of 100 cases were selected to model the morphological responses. Figure 7 shows the changes in volume with varying facAs at the same facSk. Overall, an increase in facAs led to an increase in the beach volume when facSk remained constant. This reaffirmed that an increase in the impact of wave asymmetry led to an increase in sediment transport to the land. Similar to facua, the skill value generally showed a sharp decline when it exceeded 0.33. Figure 8 presents a graph depicting the changes in the volume and skill value with varying facSk values for the same facAs. The graph exhibits irregular patterns, which are different from those of facua or facAs. An increase in facSk did not uniformly increase the volume; instead, an increase in facAs generally led to an increase in the overall volume. The increase in facSk was found to enhance sediment movement, but did not determine the direction of movement, indicating potential transport both offshore and onshore.

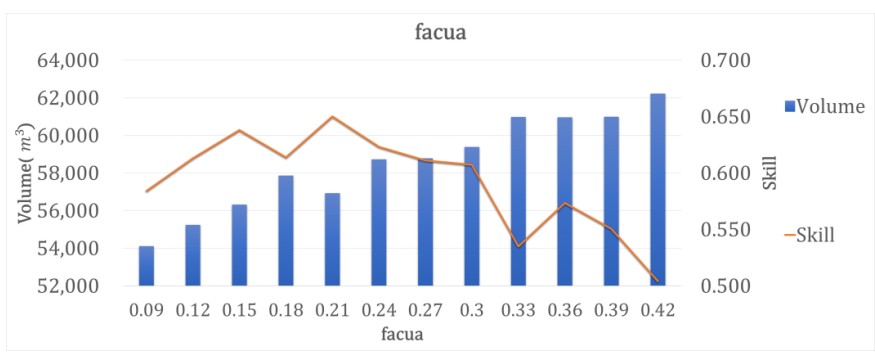

**Figure 6.** The variations in volume and skill values in response to the changes in facua.

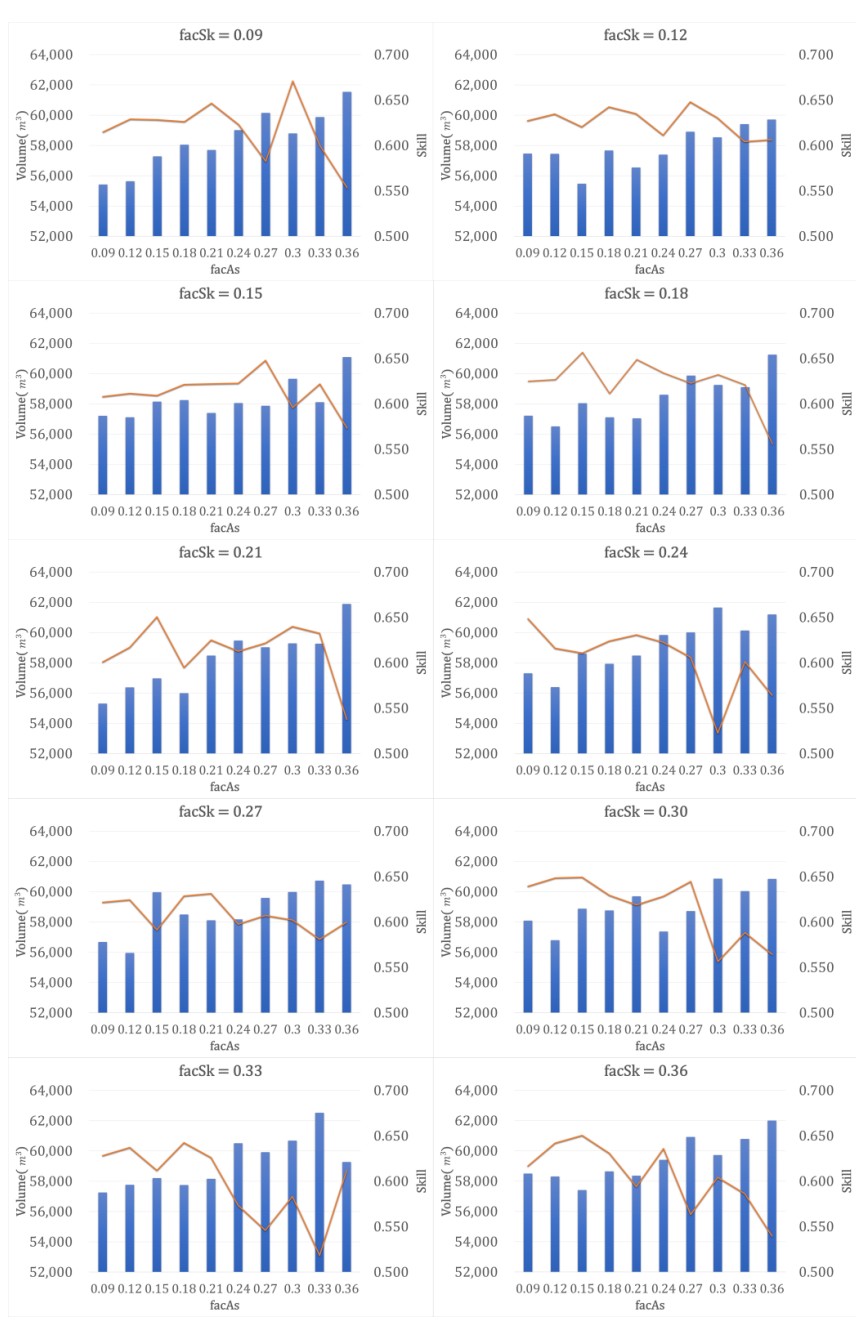

**Figure 7.** The variations in volume and skill values in response to changes in facAs, with a constant facSk.

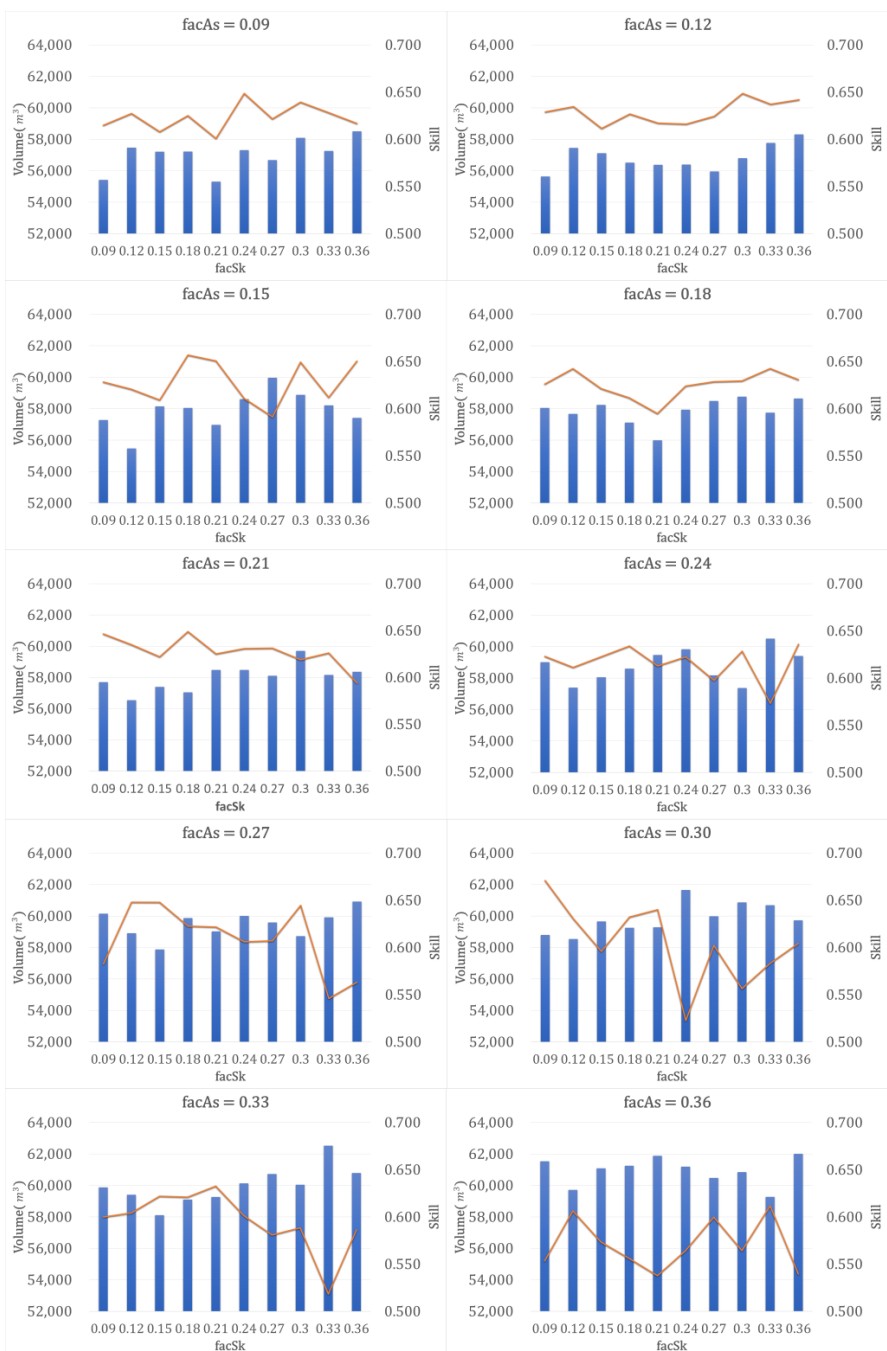

**Figure 8.** Changes in volume and skill values are shown in response to variations in facSk while maintaining a constant facAs.

## 4.2. Parameter Calibration Result: Skill and Bias

In addition to the volumetric results described in Section 4.1, skill and bias were utilized to assess the accuracy and understand the trends of the skill values for each case. The changes in skill values corresponding to variations in the parameters of facua, facAs, and facSk are shown in Figures 6–8. While generally maintaining high values of approximately 0.5, facua and facAs exhibited a sharp decline in skill values beyond 0.33. This decrease from values above 0.5 indicates the successful simulation of erosion at the front; however, erosion is underestimated when the values exceed 0.33. In contrast, facSk showed irregular patterns for skill values. The calibrated results based on these findings are summarized in Table 5. ID1 represents the case where facua is used, whereas ID2

indicates the case where facua is separated into facAs and facSk. The highest skill value, showing a volume similar to that of the observed data, was achieved by facua at 0.3, with a skill value of 0.555 and a bias of −0.083, suggesting a slight overestimation of erosion. Additionally, skill values and biases were analyzed by segmenting them into different areas. The segmentation was based on morphological changes, categorized as type A for areas with significant overall erosion, type B for deposition due to overwash, and type C for erosion at the front with minimal changes in the rear (Figure 4b). The skill values and biases of each segment are listed in Table 6. ID1, the result of the calibration using facua, showed high values overall, except for area A. Area A, characterized by severe erosion at the rear, was insufficiently simulated. Despite the inadequate simulation of the deposition at the rear, a high skill value of 0.773 was obtained for area B, indicating the result of the simulation of erosion at the front. A depiction of the elevation difference of the beach is shown in Figure 9c. Although the erosion at the front was somewhat overestimated, a high-level simulation of the front erosion was achieved compared with the observed values (Figure 9a). However, the simulation of deposition due to overwashing did not yield sufficiently reasonable results.

**Table 5.** Calibration parameters of facua (ID1), facAs, and facSk (ID2).

| ID | Waveform Parameter | | | Skill | Bias | Volume |
|---|---|---|---|---|---|---|
| | **facua** | **facAs** | **facSk** | | | |
| 1 | 0.3 | | | 0.555 | −0.083 | 59,646.9 |
| 2 | | 0.27 | 0.18 | 0.623 | −0.082 | 59,870.5 |
| Obs | | | | | | 60,406.2 |

**Table 6.** Skill and bias values for each section.

| ID | **Skill_A** | **Skill_B1** | **Skill_C** | **Skill_B2** |
|---|---|---|---|---|
| 1 | 0.261 | 0.773 | 0.665 | 0.663 |
| 2 | 0.278 | 0.850 | 0.767 | 0.676 |

| ID | **Bias_A** | **Bias_B1** | **Bias_C** | **Bias_B2** |
|---|---|---|---|---|
| 1 | 0.048 | −0.026 | −0.005 | −0.090 |
| 2 | 0.027 | −0.019 | −0.002 | −0.084 |

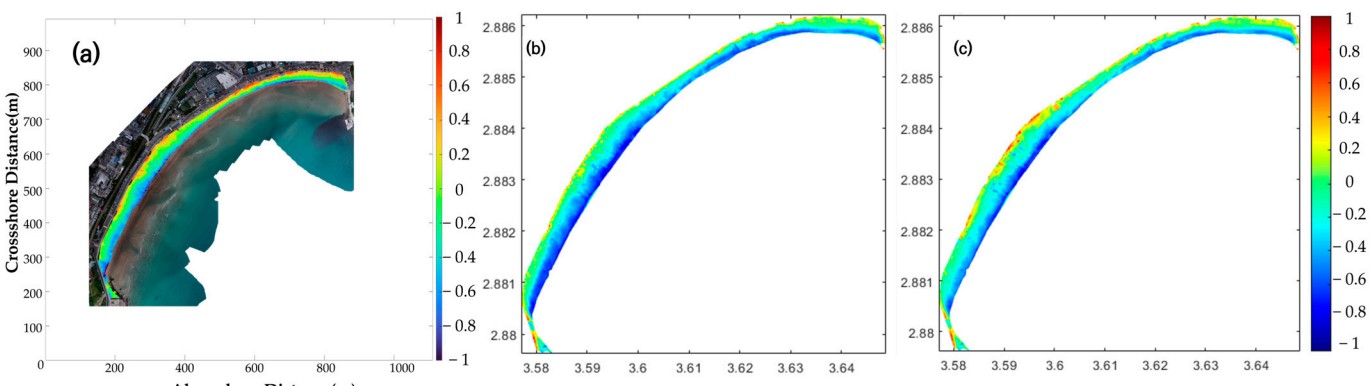

**Figure 9.** Elevation differences of observed data (**a**), ID2 (**b**), and ID1 (**c**).

By contrast, ID2, which was calibrated separately, yielded more effective results. The calibration values for facAs and facSk were 0.27 and 0.18, respectively, resulting in a volume similar to the observed data with a skill value of 0.623 and a bias of −0.082 (Table 5). These values were higher than those obtained for the facua calibration, indicating greater precision. A detailed analysis conducted by segmenting the different areas is presented in Table 6.

The skill value and bias of ID2 demonstrate that the skill in area B1 increases to 0.850, as shown in Figure 9b. Compared with Figure 9c, this figure reveals that the deposition near the revetment at the rear is more effectively simulated. Skill_A, representing areas with overall erosion, still shows low values, which can be attributed to the focus on effectively simulating deposition due to overwash during the calibration process. This resulted in a somewhat improved skill value from a separate calibration, but it remained lower than that of skill_B1. By separating facua into facAs, which increased sediment transport to the land, and facSk, which facilitated sediment movement, a more detailed calibration was possible. This allowed the simulation of the overwash and its associated deposition. Although an increase in parameters requiring calibrations could impose additional burdens on model users, it appeared necessary to simulate the overwash and resultant deposition effectively.

## 5. Discussions

Significant erosion and overwash-induced deposition around revetments were observed following the 2022 storm Hinnamnor. Such overwash events, which can directly or indirectly harm humans, underscore the importance of numerically simulating and preventing them. To simulate these morphological responses, numerical simulations based on UAV data collected before and after the typhoons were conducted. The model employed for the simulation was XBeach, a process-based, depth-averaged model specialized for short-term localized morphological changes in beaches. XBeach can simulate various hydrodynamic and morphodynamic processes and embed approximately 250 parameters that require calibrations according to the geomorphological characteristics of the target area and input data, such as wave conditions. This research utilized a parameter set from previous studies with a high BSS in the storm cluster, while calibrating the waveform-related parameters closely tied to sediment transport for a more effective overwash simulation.

The waveform parameters included facSk for skewness, facAs for asymmetry, and facua, which was an integrated representation of these two parameters. Using gamma and gamma2 from previous studies, the calibration of these three waveform parameters was performed to simulate the overwash and resultant deposition. The evaluation of the model employed simulation skill, applicable only when the ground-level data from UAVs or LiDAR were available, and bias was used to estimate the extent of overestimations of erosion and deposition. In addition, the advantages of UAVs in measuring beach volumes were leveraged as a metric for the parameter calibration.

Calibrations using facua, facAs, and facSk demonstrated an efficient simulation of frontal erosion. Both parameters had high values and skill values above 0.5, maintaining reasonable volumes while simulating frontal erosion. However, the facua fell short in simulating the deposition in the rear areas due to overwash. The increase in facua led to an overall increase in onshore sediment transport. However, this also resulted in difficulties in simultaneously simulating erosion at the front and deposition at the rear. Calibrations were performed using the facAs and facSk separation, which successfully represented both frontal erosion and rear deposition. However, the simulations of areas experiencing overall erosion were not successful. Various reasons have been hypothesized for this, with the median grain size ($D_{50}$) being a likely factor. The study area, Songjeong Beach, is predominantly sandy and contains gravel and rocky formations in its SW region. The simulation used a consistent sand grain size based on $D_{50}$, which could have contributed to these discrepancies. Another factor considered was the use of waveform-related parameters for the calibrations. The primary objective of this study was to simulate overwash and its deposition; hence, only waveform parameters directly related to sediment transport were calibrated, whereas other parameters were adopted from previous studies. Given the variety of XBeach parameters, further research involving additional calibrations is warranted.

An additional analysis was conducted to investigate the sensitivity of the volume to facua, facSk, and facAs. The facua parameter demonstrated an almost perfectly proportional relationship to the beach volume, indicating that an increase in facua activated

onshore sediment transport. Similarly, with a constant facSk, an increase in facAs led to an increase in the beach volume.

## 6. Conclusions

The following conclusions were drawn:

1.  Utilizing the parameters of facua, facSk (skewness), and facAs (asymmetry), which integrated the wave-shape parameters closely related to sediment movement, in addition to the previously proposed parameters, was effective for modeling the phenomenon of overwash. The use of facSk and facAs was particularly effective in simulating overwashing.
2.  Although an increase in the number of parameters to be calibrated could pose a burden on the model user, it was necessary to calibrate them separately when aiming for a more accurate simulation of the overwash and subsequent sediment deposition.
3.  Generally, an increase in facAs was associated with an overall increase in the beach volume (indicating increased onshore sediment transport), whereas facSk did not show a similar trend.

**Author Contributions:** Conceptualization, B.H. and K.D.; data curation, B.H. and S.C.; formal analysis, B.H.; funding acquisition, K.D.; investigation, B.H. and K.D.; methodology, B.H. and K.D.; project administration, K.D.; resources, B.H. and S.C.; software, B.H. and K.D.; supervision, K.D.; validation, B.H. and K.D.; visualization, B.H.; writing—original draft, B.H. and K.D.; writing—review and editing, B.H., K.D. and S.C. All authors have read and agreed to the published version of the manuscript.

**Funding:** This research was funded by the Korea Institute of Marine Science and Technology Promotion, the project titled 'Cyclic Adaptive Coastal Erosion Management Technology Development' (RS-2023-00256687), and the National Research Foundation of Korea (NRF-2022R1I1A3065599).

**Institutional Review Board Statement:** Not applicable.

**Informed Consent Statement:** Not applicable.

**Data Availability Statement:** Data are contained within the article.

**Conflicts of Interest:** Author Sungyeol Chang was employed by the company Haeyeon Engineering and Consultants Corporation. The remaining authors declare that the research was conducted in the absence of any commercial or financial relationships that could be construed as a potential conflict of interest.

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
