# Peer review of "Morphological Changes in Storm Hinnamnor and the Numerical Modeling of Overwash"

_jmse, doi:10.3390/jmse12010196_

Round 1
Reviewer 1 Report
Comments and Suggestions for Authors
This manuscript entitled “Morphological changes of Storm Hinnamnor and Numerical 2 modeling of Overwash” adopts XBeach model combined with UAV survey to simulate subaerial beach morphological changes pre- and post-storm and overwash phenomena. The authors focus on calibration of waveform parameters to effectively model overwash and resultant sediment deposition, and assess the model's performance and its sensitivity to beach volume changes. Overall, the manuscript is written well, but there are a few minor issues need to be checked or improved.
1. Line 78-79, change “wave-dominate area” to “wave-dominated area”.
2. Line 101-102, “westerly waves, 90-100 degrees”, in which the waves with an azimuth angle of 90-100 degrees should be easterly waves.
3. Please check the azimuth of erosion described in the two sentences: Line 131, “The most extreme erosion occurred in the northeastern NE region of the beach”, and Line 153-154, “Evident erosion occurred both in the frontal and rear sections when observed from the NE direction”.
4. Figure 5 seems to display only the computational domain, not the grid size.
5. The quality of Figure 6 and Figure 7 needs to be improved, especially the font size.
6. Some of the references are incomplete, such as [19], [24] etc.
Author Response
Firstly, thank you for taking the time to write a review.
We have reviewed the areas you highlighted as needing modification and would like to inform you of the changes made.
Details of these revisions have been organized in the attached file.
Thank you very much for your guidance.

Reviewer 2 Report
Comments and Suggestions for Authors
See attached file.

See attached file.
Author Response

(The authors gave the same response as above.)
